# Mid- and Long-Term Atrio-Ventricular Functional Changes in Children after Recovery from COVID-19

**DOI:** 10.3390/jcm12010186

**Published:** 2022-12-26

**Authors:** Jolanda Sabatino, Costanza Di Chiara, Angela Di Candia, Domenico Sirico, Daniele Donà, Jennifer Fumanelli, Alessia Basso, Pietro Pogacnik, Elena Cuppini, Letizia Rosa Romano, Biagio Castaldi, Elena Reffo, Alessia Cerutti, Roberta Biffanti, Sandra Cozzani, Carlo Giaquinto, Giovanni Di Salvo

**Affiliations:** 1Pediatric and Congenital Cardiology Unit, Department for Women’s and Children’s Health, University Hospital of Padova, Via Nicolò Giustiniani 2, 35128 Padova, Italy; 2Department for Women’s and Children’s Health, Division of Pediatric Infectious Diseases, University Hospital of Padova, Via Nicolò Giustiniani 2, 35128 Padova, Italy; 3Department of Medical and Surgical Sciences, Division of Cardiology, Magna Graecia University of Catanzaro, Viale Europa 1, 88100 Catanzaro, Italy

**Keywords:** long COVID-19, children, speckle-tracking analysis

## Abstract

Background: Although most children may experience mild to moderate symptoms and do not require hospitalization, there are little data on cardiac involvement in COVID-19. However, cardiac involvement is accurately demonstrated in children with MISC. The objective of this study was to evaluate cardiac mechanics in previously healthy children who recovered from severe acute respiratory syndrome coronavirus-2 (SARS-CoV-2) infection in a long-term follow-up by means of two-dimensional speckle-tracking echocardiography (STE). Methods: We analyzed a cohort of 157 paediatric patients, mean age 7.7 ± 4.5 years (age range 0.3–18 years), who had a laboratory-confirmed diagnosis of SARS-CoV-2 infection and were asymptomatic or mildly symptomatic for COVID-19. Patients underwent a standard transthoracic echocardiogram and STE at an average time of 148 ± 68 days after diagnosis and were divided in three follow-up groups (<180 days, 180–240 days, >240 days). Patients were compared with 107 (41 females—38%) age- and BSA-comparable healthy controls (CTRL). Results: Left ventricular (LV) global longitudinal strain (post-COVID-19: −20.5 ± 2.9%; CTRL: −21.8 ± 1.7%; *p* < 0.001) was significantly reduced in cases compared with CTRLs. No significant differences were seen among the three follow-up groups (*p* = NS). Moreover, regional longitudinal strain was significantly reduced in LV apical-wall segments of children with disease onset during the second wave of the COVID-19 pandemic compared to the first wave (second wave: −20.2 ± 2.6%; first wave: −21.2 ± 3.4%; *p* = 0.048). Finally, peak left atrial systolic strain was within the normal range in the post-COVID-19 group with no significant differences compared to CTRLs. Conclusions: Our study demonstrated for the first time the persistence of LV myocardial deformation abnormalities in previously healthy children with an asymptomatic or mildly symptomatic (WHO stages 0 or 1) COVID-19 course after an average follow-up of 148 ± 68 days. A more significant involvement was found in children affected during the second wave. These findings imply that subclinical LV dysfunction may also be a typical characteristic of COVID-19 infection in children and are concerning given the predictive value of LV longitudinal strain in the general population.

## 1. Introduction

SARS-CoV-2 infection was initially perceived as a mild disease amongst paediatrics. However, it is now widely recognized that the spectrum of documented COVID-19 cases varies from asymptomatic to severe, with cases of hyper-inflammatory response, known as multisystem inflammatory syndrome in children (MIS-C). MISC is sometimes observed 3–4 weeks after the acute phase, with longer-term consequences [1,2,3].

Although most children may experience mild to moderate symptoms and do not require hospitalization, data [4] on cardiac consequences during COVID-19 acute infection are scarce. On the other hand, cardiac sequelae, in terms of reduced left ventricular function, have recently been well documented in children who developed MISC [5,6].

Although COVID-19 may have some degree of myocardial inflammation, standard echocardiographic evaluations of LV systolic function often fail to identify this. Speckle tracking echocardiography (STE) has emerged in the past two decades as an effective tool to detect subclinical alterations of myocardial function, while assessing myocardial deformation (or strain) with a good sensitivity [7,8].

A recent longitudinal speckle-tracking echocardiography (STE) evaluation of 80 adult patients [9], until three months after hospitalization from COVID-19, showed that a quarter of patients still exhibited LV systolic dysfunction based on STE normal ranges. Additionally, in the same cohort, LV STE did not significantly improve during follow-up, implying subclinical LV dysfunction may be a typical characteristic of recovering from COVID-19 infection.

Therefore, our aim was to evaluate left ventricular and atrial myocardial deformation properties using STE parameters in previously healthy paediatric patients recovering from COVID-19 infection after a mid- and long-term follow-up.

## 2. Materials and Methods

### 2.1. Study Design and Population

The study was designed as a single-centre observational prospective investigation of Italian children who tested positive for SARS-CoV-2 infection between 17 February 2020 and 15 December 2021. The project was realized by the ‘COVID-19 Family Cluster Follow-up Outpatient Clinic’ (CovFC) of the Department for Women’s and Children’s Health (W&CHD) of the University of Padua, Italy. The Institutional Review Board (IRB) approved the study protocol, and all the children’s parents provided written informed consent.

Subjects enrolled in the study were 0–18 years old and chosen from households with at least one diagnosed case of COVID-19 in the family. Families were enrolled four or more weeks after infection, after a referral from the family paediatrician or the Paediatric COVID-19 unit of the Department of Women’s and Children’s Health of Padua, on the basis of the following inclusion criteria: (a) children of paediatric age (<18 years) with laboratory-confirmed SARS-CoV-2 infection; and (b) one or more family member/s with a history of COVID-19. During enrolment, data on demographic features and past medical history, symptoms during COVID-19 and post-infection were collected, and a clinical evaluation was performed.

### 2.2. Data Collection and Definitions

Subjects who tested positive for SARS-CoV-2 RT-PCR test or showed serological evidence of previous exposure to the virus were considered “Confirmed cases of COVID-19”. For each COVID-19 case, a baseline date was defined as follows: (1) for symptomatic cases: the first date between the onset of symptoms or the date of the first positive SARS-CoV-2 molecular assay; (2) for asymptomatic cases: the date of the first positive molecular assay or, in those with only serologically confirmed COVID-19 and with negative/undetermined nasal-pharyngeal swab (NPs), by the family outbreak temporal sequence, coinciding with the date of symptoms onset in the family cluster.

Data collection included demographical details (such as gender, date of birth and BSA), comorbidities, vaccination record, COVID-19 course (time of infection, symptoms, and medical treatment) and follow-up measures (serological assays and heart tests, such as ECG and TTE). All data were collected, maintaining confidentiality, and were anonymized for statistical analysis.

Severity of COVID-19 was classified as mild, moderate, severe, or critical, according to the WHO classification based on clinical features, laboratory tests and chest radiograph imaging where available [10].

The subjects who tested negative during the aforementioned diagnostic tests or had an acute course with moderate or severe symptoms (WHO ≥ 2) were excluded from the study.

Two periods of time or “COVID-19 waves” were identified in our study population and defined as follows: a first wave occurring from 17 February to 18 September 2020 and a second wave from 19 September 2020 to 18 February 2021.

### 2.3. Control Group

One hundred and seven controls of comparable age and BSA to the confirmed cases of COVID-19 (Table 1) were subsequently selected by the Paediatric Cardiology Unit of the W&CHD of the University Hospital of Padua, Italy. Controls were recruited among healthy subjects admitted for atypical chest pain or innocent murmur, who were not on drug therapy. Their cardiac evaluation, ECG, and transthoracic echocardiogram (TTE) were all normal. Subjects (study group and control group) with two-dimensional (2D) TTE views not optimal for longitudinal strain analysis (more than 2 LV segments not visualized) were excluded.

### 2.4. Serological Assays

Quantification of anti-SARS-CoV-2 S-RBD IgG Ab was performed through commercially available chemiluminescent assays (CLIA) (Snibe Diagnostics, New Industries Biomedical Engineering Co., Ltd. [Snibe], Shenzhen, China). This method, previously validated elsewhere [11], quantitatively determines the IgG antibodies to RBD portion of SARS-CoV-2 spike protein. All analyses were conducted on MAGLUMI™ 2000 Plus (Snibe Diagnostics), with results expressed in the kilo binding antibody unit (kBAU/L), in accordance with WHO International Standard for anti-SARS-CoV-2 immunoglobulin. Samples recording titres >4.33 kBAU/L were considered positive.

A high-throughput method for the plaque reduction neutralization test (PRNT) was used to quantify NAbs in serum samples for a subgroup of patients infected by SARS-CoV-2 within the first and second waves [12]. The neutralization titre was defined as the reciprocal of the highest dilution resulting in a reduction of >50% in the control plaque count (PRNT50). Samples recording titres equal to or above 1:10 were considered positive. 

### 2.5. Cardiac Evaluation

COVID-19-positive children were subjected to standard cardiac assessment for an average time of 148 ± 68 days after baseline. ECG and TTE were also performed using the GE Vivid E9 Ultrasound System (GE Healthcare, Horten, Norway) in compliance with the most recent recommendations [3,4]. Left ventricular ejection fraction (LVEF) by TTE was calculated by means of the Simpson method. Left ventricular longitudinal strain analysis was performed via 2D Speckle tracking echocardiography (STE), using the GE EchoPac Software offline (GE Healthcare, V.202, Horten, Norway). 

Our standard protocol for STE has been described in previous reports [13,14,15,16]. To ensure a precise and consistent STE evaluation, the clearest frames of the apical echocardiographic projections of the heart were selected on the GE EchoPac Software. The left atrium’s performance was assessed using four- and two-chamber views, while the left ventricle was examined in two-, three- and four-chamber views. Three points were selected on the frames: two annular and one on the atrial roof for the LA, two annular and one apical for the LV. Subsequently, the software semi-automatically identified the chamber wall’s contour, then manually corrected this by carefully following the endocardial margins [15]. By tracking the movement of the myocardium during the cardiac cycle, the GE EchoPac’s algorithm ultimately calculated the global longitudinal strain (GLS) and the individual segmental longitudinal strains (SLS) of the walls of the heart.

Analysis of the standard TTE and STE was performed by an experienced echocardiographer blind to the clinical data.

### 2.6. Reproducibility 

Inter-rater and intra-rater reliability were evaluated using the intraclass correlation coefficients as previously described [16].

### 2.7. Statistical Analysis

Categorical variables are shown as a percentage (%), while continuous variables are shown as mean ± standard deviation. The Shapiro–Wilk test and visual inspection of histograms were used to evaluate normality of variables. Normally distributed continuous variables were tested using Student’s *t*-tests, while the non-parametric variables were tested using the Mann–Whitney U-test. The Chi-square test was applied for categorical variables. The Bonferroni correction test was performed for multiple hypothesis testing, to check for false positives. A *p*-value < 0.025 was considered statistically significant.

## 3. Results

### 3.1. Baseline Characteristics

Among the patients admitted to our institution from 1 March 2020 to 15 December 2021, we prospectively included 157 (62 females, 39%) consecutive previously healthy children who had an asymptomatic or mildly symptomatic COVID-19 (WHO stages 0 or 1) at least eight weeks before.

All included children were previously healthy, without any evidence of former cardiac pathologies.

Patients were compared with 107 (41 females—38%) age- and BSA-comparable healthy controls (CTRL).

Baseline and echocardiographic variables of the study population are displayed in Table 1.

Among the post-COVID-19 cases, 35 patients (22%) had asymptomatic COVID-19 (WHO = 0), while the other 122 (78%) experienced mild symptoms (WHO = 1) during the acute phase of the infection. 

All the cardiac examinations were performed after an average follow-up of 148 ± 68 days from disease onset.

### 3.2. Standard Echocardiographic Measurements

LV dimensions, global systolic and diastolic function, and left atrial volume indexed to BSA (LAVi) were comparable in the post-COVID-19 and CTRL groups (Table 1).

No hints of relevant coronary artery dilation (Z scores all <2) or pericardial effusion were significant among post-COVID-19 cases. 

Finally, the right ventricular longitudinal function, calculated with the tricuspid annular plane systolic excursion (TAPSE) method, was similar between the groups (19.8 ± 3.0 mm vs. 20.1 ± 3.4 mm; *p* = 0.822).

### 3.3. Left Ventricular Longitudinal Strain

Global longitudinal strain (post-COVID-19: −20.5 ± 2.9%; CTRL: −21.8 ± 1.7%; *p* < 0.001) was significantly reduced in children belonging to the post-COVID-19 group compared to controls, although still within the normal range (Figure 1A). 

Furthermore, LV strain regional analysis showed a significant reduction in the post-COVID-19 group (Figure 1B and Figure 2A,B).

Thirty-seven post-COVID-19 children reached a follow-up of >240 days since the acute onset of the disease, with a mean GLS of −20.1 ± 2.8%. Overall, 28 children were examined at a follow-up of 180–240 days with a mean GLS of −20.6 ± 4.4%, and the remaining 92 children were seen before 180 days of follow-up, with a GLS of −20.7 ± 2.4%. No significant differences were seen among those three follow-up groups (*p* = NS) (Figure 3A).

Eleven (7%) post-COVID-19 cases had impaired GLS values <−16%, while in 95 post-COVID-19 cases (60%), a regional strain was present in <−16% in more than two LV segments.

Those patients did not show any significant difference compared to the remaining children, with regard to BSA and age.

None of the CTRLs showed a GLS or more than two LV segments with a strain value <−16%.

We did not observe any significant correlations between serological reports (IgM, IgG, and PRNT) and LS impairment (*p* = 0.632, *p* = 0.308 and *p* = 0.112, respectively).

ECG displayed a sinus rhythm in all groups of post-COVID-19 children. Five patients (3%) showed a slightly abnormal ECG (four cases with repolarization anomalies and one patient with sinus bradycardia). However, they did not exhibit any significant variation in LS values compared with the rest of the population.

### 3.4. Left Ventricular Longitudinal Strain According to Different Pandemic Waves

Interestingly, LV strain segmental analysis revealed a significant strain reduction in the LV apical-wall segments (second wave: −22.0 ± 4.4%; first wave: −24.4 ± 5.4%; *p* = 0.006) among cases with disease onset, which occurred during the second wave of the pandemic, compared to the first wave.

According to the two different pandemic waves, no other differences in age, BSA, LV dimensions, peak systolic left atrial strain were observed.

### 3.5. Left Atrial Reservoir Strain

Peak left atrial systolic strain values were within the normal range in the post-COVID-19 group, and we found no significant differences compared to CTRL (post-COVID-19: 49.1 ± 12%; CTRL: 49.5 ± 18%) (Figure 4).

Reproducibility analysis. Indices of intra- and inter-observer variability were very good for longitudinal strains (7 ± 7%, ICC: 0.92 and 7 ± 8%, ICC: 0.9, respectively).

## 4. Discussion

Our findings shed new light on the cardiac impact of COVID-19 in paediatric age. In fact, we showed for the first time that 60% of children who recovered from asymptomatic or mildly symptomatic COVID-19 still exhibit mild subclinical systolic cardiac impairment after an average follow-up of 148 ± 68 days from disease onset. This subtle impairment in myocardial deformation was worse in the LV apical region of COVID-19 children who recovered during the second wave compared to the first wave.

While cardiac involvement is well recognized in MISC, there is still a lack of knowledge about cardiovascular consequences in children with a mildly symptomatic COVID-19 course.

A study from our group [6] recently demonstrated an abnormal left ventricular deformation in a cohort of 53 children at three months of follow-up after asymptomatic or mildly symptomatic COVID-19. The present study completes and extends these results by evaluating the ventricular and atrial deformation in a larger cohort (157 children) with a longer-term follow-up (average follow-up of 148 ± 68 days since COVID-19 disease’s onset), demonstrating the persistency of these myocardial abnormalities and differences in terms of cardiac involvement in the two COVID-19 waves. Our findings imply that subclinical LV dysfunction may also be a typical characteristic of COVID-19 infection in children.

Underlining reasons for lasting myocardial consequences after acute illnesses have not been successfully explained [17,18,19,20,21,22,23,24,25,26]. A possible explanation is the persistency of viral reservoirs, which may evoke a chronic inflammatory response in the heart. 

Another possible mechanism for enduring cardiac damage may be an autoimmune response to cardiac self-antigens.

Our findings suggest that the extent of the immune system response to acute infection may not be related to the subclinical enduring cardiac damage in the mid- and long-term follow-up. Therefore, the virus, per se, could directly cause these persistent myocardial deformation abnormalities.

A recent systematic review of pathology-derived cardiac changes in patients with COVID-19, which included 50 studies with more than 500 cases, demonstrated that most prevalent chronic changes were myocardial hypertrophy and fibrosis and confirmed the significant prevalence of acute and chronic cardiac damages in COVID-19 and the SARS-CoV-2 cardiac tropism [27].

Moreover, in line with our results of subtle myocardial impairment, a recent report showed a direct cardiac involvement associated with intramyocardial inflammation and a SARS-CoV-2 genome positivity in in endomyocardial biopsies [28].

Our data suggest that children infected during the second wave had a more pronounced LV myocardial involvement. However, since we did not have data on the specific SARS-CoV-2 variants involved, this observation is merely epidemiological. 

The persistence of GLS and regional LV longitudinal strain abnormalities in children after COVID-19 infection is a particular concern, since even in low-risk population studies [29], or in presence of a normal LV ejection fraction [30], a reduced GLS is associated with higher morbidity and mortality. The pathophysiologic mechanisms behind the changes in myocardial longitudinal strain remain unclear, but its role in diagnosis and possible future treatment strategies is significant.

## 5. Conclusions

For the first time, our study demonstrated the persistence of LV myocardial deformation abnormalities after an average follow-up of 148 ± 68 days from an asymptomatic or mildly symptomatic COVID-19 course (WHO stages 0 or 1) in previously healthy children. Myocardial alterations were more pronounced during the second wave of COVID-19. These findings imply that subclinical LV dysfunction may complicate even asymptomatic or mildly symptomatic children affected by COVID-19, with worrisome implications given the predictive value of LV longitudinal strain.

## Figures and Tables

**Figure 1 jcm-12-00186-f001:**
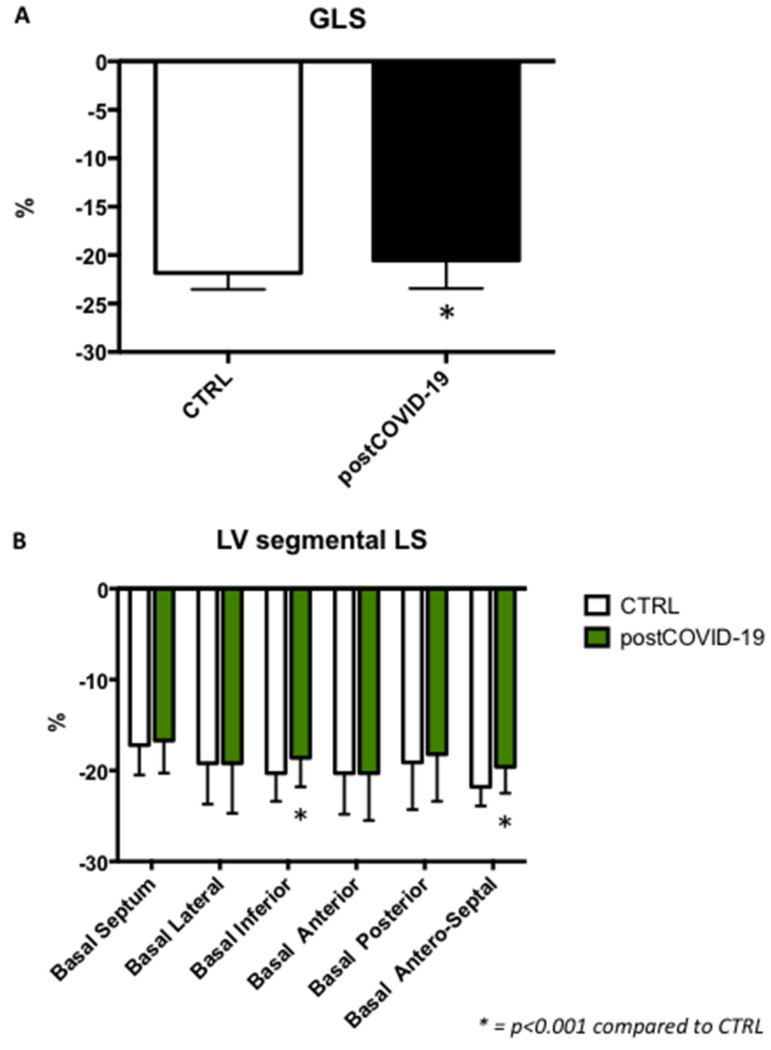
Global longitudinal strain in children belonging to the post-COVID-19 group compared to controls (**A**). Basal segmental analysis in post-COVID-19 group compared to controls (**B**).

**Figure 2 jcm-12-00186-f002:**
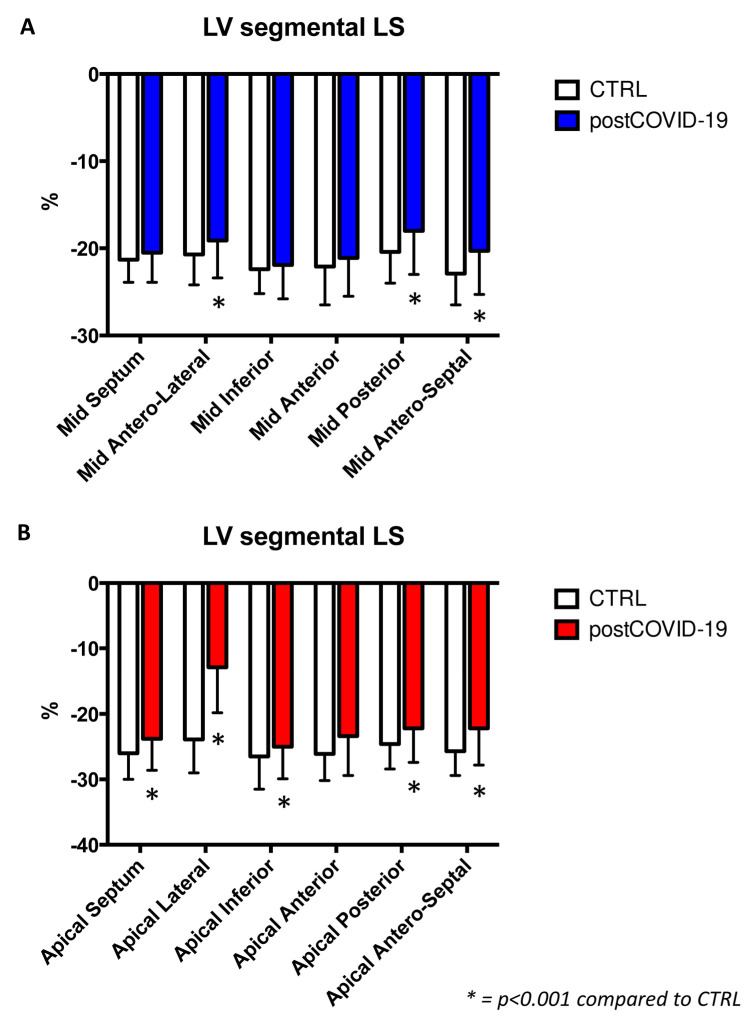
The mid (**A**) and apical (**B**) LV segmental analysis illustrated in detail.

**Figure 3 jcm-12-00186-f003:**
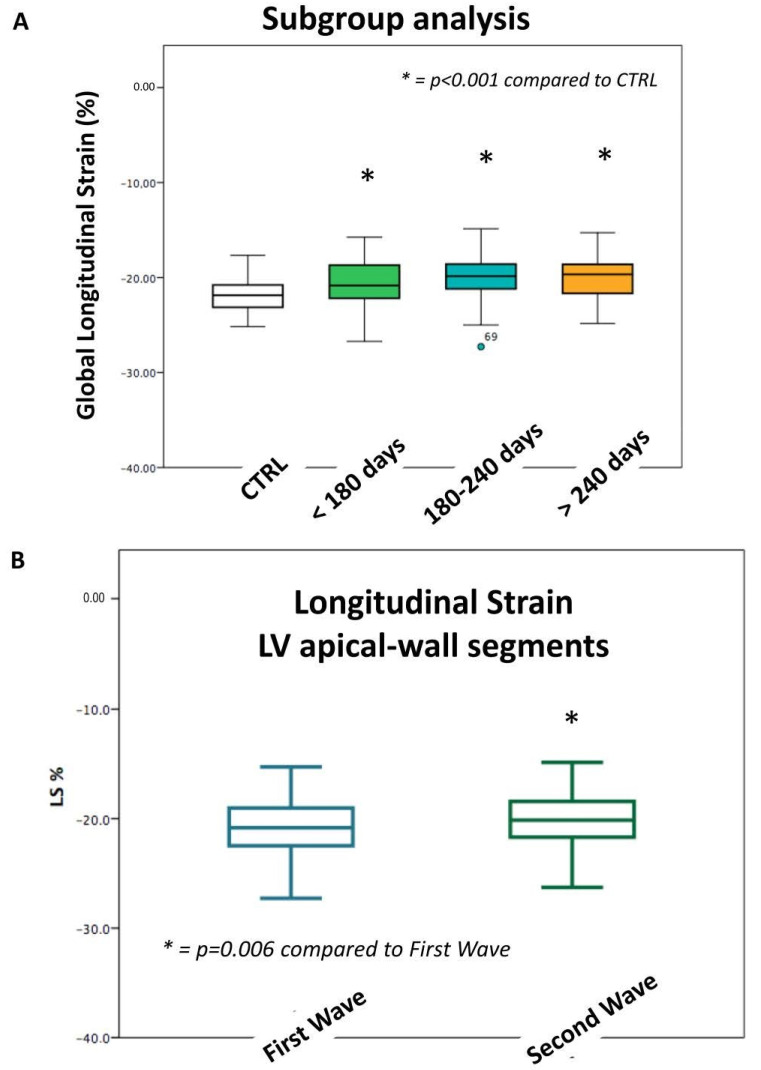
Longitudinal strain of the post-COVID-19 group according to different follow-up (**A**) and different pandemic waves (**B**).

**Figure 4 jcm-12-00186-f004:**
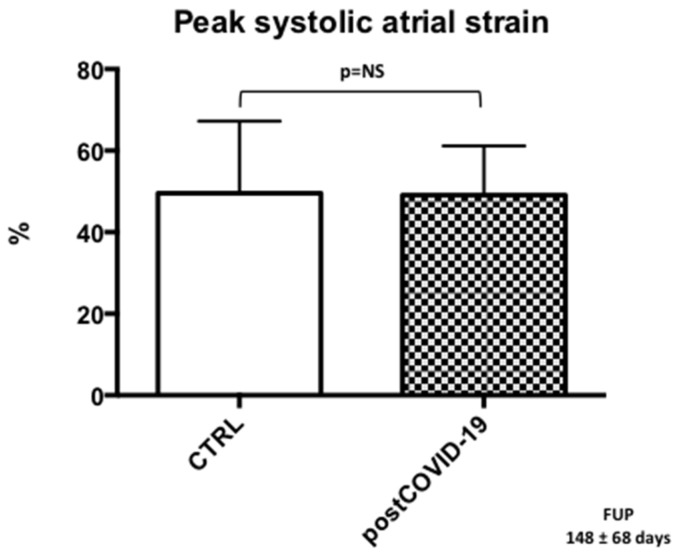
Peak systolic left atrial strain in the post-COVID-19 group compared with CTRL.

**Table 1 jcm-12-00186-t001:** Groups.

Clinical and Echocardiographic Variables	Post-COVID-19 (N = 157)	CTRL (N = 107)
Age (yrs)	7.7 ± 4.5	11.0 ± 4.3
Age range (yrs)	0.3–18	0.3–18
Female, *n* (%)	62 (39)	41 (38)
Body Surface Area (BSA) (m^2^)	1.2 ± 0.4	1.3 ± 0.4
LVEDD (mm)	39.1 ± 6.7	40.7 ± 6.3
LVEDD Z score	−0.33 ± 0.91	−0.23 ± 0.75
LVESD (mm)	24.8 ± 4.7	26.2 ± 4.4
LVESD Z score	−0.16 ± 1.06	−0.15 ± 0.82
LVEF (%)	65.6 ± 4	65.0 ± 5
LAVi (mL/m^2^)	17.3 ± 6.4	17.7 ± 4.9
TAPSE (mm)	19.8 ±3.0	20.1 ±3.4
E/A ratio	1.9 ± 0.51	1.8 ± 0.51
E/E’ avg, ratio	5.9 ± 1.1	6.2 ± 1.3

Values are mean ± SD, or *n* (%). BSA: body surface area. LVEDD: left ventricular end diastolic diameter. LVESD: left ventricular end systolic diameter. LVEF: left ventricular ejection fraction. LAVi: left atrial volume indexed to BSA. TAPSE: tricuspid annular plane systolic excursion.

## Data Availability

Data is unavailable due to privacy and ethical restrictions.

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
