# Peer review of "Mid- and Long-Term Atrio-Ventricular Functional Changes in Children after Recovery from COVID-19"

_jcm, 2022, doi:10.3390/jcm12010186_

Round 1

Reviewer 2 Report

Sabatino et al. performed echocardiography (including speckle-tracking derived deformation analysis) in a cohort of 157 consecutive paediatric patients with  COVID-19, who were asymptomatic or only mildly affected during the acute infection. Echo assessment took place about 150 days after initial diagnosis and was compared to a matched group of 107 healthy controls. The authors could demonstrate that even a mild COVID-19-infection resulted in a significantly reduced LV GLS in comparison to the healthy cohort.

The authors have to be congratulated for the well conducted study and the highly relevant results in consequence. The prospective enrolment of asymptomatic to mildy affectes children by identifying families with at least one diseased family member is a very reasonable approach. Data collection and image acquisition was conducted carefully and according to current recommendations. 

To further clarify the results, following information should be provided:

- Was there a correlation between disease severity (completely asymptomatic vs. mildy affected) and strain reduction?

- How many subjects belonged to wave 1 vs. wave 2? Besides regional strain differences, how was GLS in those two subgroups?

- Was there a correlation between serological disease burden (titer) and strain reduction?

- Were there any abnormalities concerning ECG or clinical cardiac assessment?

Reviewer 3 Report

I read the paper with GREAT interest. Sequelae of COVID in adults and children are big topics now. Methodology is perfect. The English writing is GREAT too. one detail in line 66: All patients provided written informed consent. Maybe this is the way for pediatric publication. As adult cardiologist, I understand that children cannot sign consent. Should we write: Consent was given by parents?   

Round 2

Reviewer 1 Report

I have no futher comments.